# Understanding heterogeneous mechanisms of heart failure with preserved ejection fraction through cardiorenal mathematical modeling

**Sanchita Basu**[1], **Hongtao Yu**[1,2], **Jonathan R. Murrow**[3], **K. Melissa Hallow**[1,4] *

**1** School of Chemical, Materials, and Biomedical Engineering, University of Georgia, Athens, Georgia, United States of America, **2** Clinical Pharmacology and Quantitative Pharmacology, Clinical Pharmacology & Safety Sciences, R&D, AstraZeneca, Gaithersburg, Maryland, United States of America, **3** Department of Cardiology, Piedmont Athens Regional Hospital, Athens, Georgia, United States of America, **4** Department of Epidemiology and Biostatistics, University of Georgia, Athens, Georgia, United States of America

\* hallowkm@uga.edu

**Data Availability Statement:** All relevant data are within the manuscript and its Supporting Information files.

## Abstract

In contrast to heart failure (HF) with reduced ejection fraction (HFrEF), effective interventions for HF with preserved ejection fraction (HFpEF) have proven elusive, in part because it is a heterogeneous syndrome with incompletely understood pathophysiology. This study utilized mathematical modeling to evaluate mechanisms distinguishing HFpEF and HFrEF. HF was defined as a state of chronically elevated left ventricle end diastolic pressure (LVEDP > 20mmHg). First, using a previously developed cardiorenal model, sensitivities of LVEDP to potential contributing mechanisms of HFpEF, including increased myocardial, arterial, or venous stiffness, slowed ventricular relaxation, reduced LV contractility, hypertension, or reduced venous capacitance, were evaluated. Elevated LV stiffness was identified as the most sensitive factor. Large LV stiffness increases alone, or milder increases combined with either decreased LV contractility, increased arterial stiffness, or hypertension, could increase LVEDP into the HF range without reducing EF. We then evaluated effects of these mechanisms on mechanical signals of cardiac outward remodeling, and tested the ability to maintain stable EF (as opposed to progressive EF decline) under two remodeling assumptions: LV passive stress-driven vs. strain-driven remodeling. While elevated LV stiffness increased LVEDP and LV wall stress, it mitigated wall strain rise for a given LVEDP. This suggests that if LV strain drives outward remodeling, a stiffer myocardium will experience less strain and less outward dilatation when additional factors such as impaired contractility, hypertension, or arterial stiffening exacerbate LVEDP, allowing EF to remain normal even at high filling pressures. Thus, HFpEF heterogeneity may result from a range of different pathologic mechanisms occurring in an already stiffened myocardium. Together, these simulations further support LV stiffening as a critical mechanism contributing to elevated cardiac filling pressures; support LV passive strain as the outward dilatation signal; offer an explanation for HFpEF heterogeneity; and provide a mechanistic explanation distinguishing between HFpEF and HFrEF.

**Funding:** K.M.H. was funded by AstraZeneca Pharmaceuticals, Inc. (www.astrazeneca.com). H.Y. is an employee of AstraZeneca and owns AstraZeneca stock or stock options. Besides H.Y. and P.G. listed in the acknowledgements, the funders had no further role in study design, data collection and analysis, decision to publish, or preparation of the manuscript.

**Competing interests:** The authors have declared that no competing interests exist.

## Author summary

Over the last three decades, treatments for heart failure with reduced ejection fraction (HFrEF) have improved survival rates, but therapies for heart failure with preserved ejection fraction (HFpEF) lack similar success. HFpEF is complex, with unclear causes. This study used computer models of the heart and kidneys to explore what sets HFpEF apart. We tested various factors that might contribute to HFpEF. We found that increased heart tissue stiffness was the most important factor. Severely elevated heart stiffness alone, or mildly elevated stiffness combined with other issues like reduced heart contractility, high blood pressure, or stiff blood vessels, could lead to the HFpEF. We also looked at how these factors affect heart remodeling. In HFrEF, the heart enlarges in response to excessive stretching when pressure inside the heart is high. If the heart tissue is stiffer, it may resist stretching even when other problems increase heart filling pressure, minimizing the enlargement signal. This may explain why HFpEF hearts do not enlarge, maintaining normal ejection fractions despite high filling pressures. By better understanding key factors that differentiate HFpEF from HFrEF, these findings could guide future treatments for this complex condition.

## Introduction

Heart failure (HF) occurs when the heart is unable to maintain sufficient cardiac output (CO) to supply the metabolic needs of the body at normal cardiac filling pressures [1], resulting in the signs and symptoms of heart failure, including dyspnea, edema, fatigue, and reduced exercise capacity. Heart failure affects more than 64 million globally [2]. It is often accompanied by reduced ejection fraction (HFrEF, EF < 40%), but about half of incident heart failure cases have a preserved ejection fraction (HFpEF, EF > 50%) [3].

Management of HFrEF has greatly improved and survival rates have increased over the past 20 years due to the advent of effective therapies [4]. However, outcomes in HFpEF remain poor, and therapies effective in HFrEF have not shown the same benefits in HFpEF clinical trials [5,6]. Sodium-glucose cotransporter-2 (SGLT2) inhibitors are the first class of drug to demonstrate clinical benefit [7,8], but the reasons for their success while others were not effective remain unclear, highlighting our limited understanding of HFpEF.

A major challenge to improving outcomes in HFpEF is that the underlying mechanisms are heterogeneous and not well understood [9]. While HFrEF patients typically have a history of ischemic heart disease and a predictable pattern of progressive ventricle dilatation, HFpEF patients often have cardiac geometries and risk factor profiles that are similar to non-HF patients with left ventricle (LV) hypertrophy [10]. However, the distinguishing difference is that they have higher cardiac filling pressures [11–13]. A subset of HFpEF patients have a history of ischemic heart disease and impaired cardiac contractility [14,15], but many have reported normal systolic function [16]. Hypertension is the most common comorbidity in HFpEF, but blood pressures in HFpEF patients are also not different from those with LV hypertrophy (LVH) [10]. Increased myocardial passive stiffness is observed in nearly all studies of HFpEF [17–19]. This stiffening was originally thought to be due to increased collagen deposition and fibrosis, but more recently, evidence has emerged that the myocytes themselves become stiffer due to hyperphosphorylation of titin filaments [20–22]. Still, increased cardiac stiffness alone does not seem to explain the signs and symptoms of HFpEF, as elevated LV stiffness is also observed in LVH patients who do not have heart failure [15,23].

Multiple other abnormalities in diastolic, systolic, and vascular function have been reported in HFpEF, but again none seem to be sufficient to explain the syndrome. Delayed ventricle relaxation during diastole is frequently but not always observed in HFpEF [24], as indicated by higher isovolumic relaxation times (IVRT) [19,25] and relaxation time constants (tau) [19]. Vascular stiffness is increased in HFpEF [17] and is strongly associated with aging and obesity–both risk factors for HFpEF [10,26–30]. Impaired ventricular-vascular coupling [31] may play a role; both arterial elastance (Ea) and ventricular end-systolic elastance (Ees) are increased in HFpEF relative to healthy controls. Venous compliance and venous capacitance are decreased in HFpEF patients relative to patients with non-cardiac dyspnea, and the severity is correlated with obesity [32]. Renal dysfunction is a co-morbidity in more than half of HFpEF patients [33,34]. However, each of these mechanisms also occur in many patients who never ultimately develop HFpEF [10,17].

While none of these mechanisms individually seem to explain HFpEF, each likely plays some role in at least a portion of the HFpEF population. However, the relative contribution of each is not known. In addition, some observed functional changes may be causal, while others may be adaptive or maladaptive consequences of other causal mechanisms. A better quantitative understanding of the relative contributions of potential mechanisms of HFpEF is needed in order to better treat and prevent this disease.

In addition to the uncertainty around causal mechanisms of HFpEF, another mystery of HFpEF is why elevated cardiac filling pressures do not cause outward remodeling and ventricle dilatation in these patients. Abnormally high filling pressure is a key distinguishing feature between patients with HFpEF and non-symptomatic subjects [11–13]. Elevated filling pressure generally is associated with volume-overload, and other conditions of volume overload (e.g. impaired systolic function, mitral regurgitation) are characterized by progressive eccentric or outward dilatation [35]. However, HFpEF is the anomaly–filling pressures are elevated, but outward remodeling is limited.

Mathematical modeling has been utilized extensively to improve our understanding of cardiac growth and remodeling laws under normal and pathologic conditions [36–41], and can be a tool for evaluating potentially multi-factorial causes of HFpEF. Most previous models treat the cardiovascular system as a closed system and do not account for the role of natriuretic control of fluid status and blood volume/pressure by the kidney. However, because elevated cardiac filling pressure is a key feature of heart failure, hemodynamic feedback control of fluid status provided by the kidney likely plays a critical role. We have previously developed a model that links cardiac and kidney function [39], and used this model to investigate pathophysiology and pharmacologic responses in HFrEF [42]. A key feature of this model, compared to close-system cardiac models that treat fluid volume as constant rather than controlled by the kidney, is that the effects of alterations in cardiac and vascular function on fluid status can be simulated mechanistically; congestion and elevated cardiac filling pressures emerge mechanistically from fluid retention as an adaptive renal response to maintain CO. Similarly, changes in afterload are simulated mechanistically, e.g. hypertension can be induced mathematically through renal mechanisms that cause sodium/water retention and/or neurohormonal feedback on peripheral resistance [43]. The effects of changes in preload and afterload on cardiac remodeling over time can then be simulated through growth and remodeling laws that link myocardial loading to changes in myocyte diameter/length, and subsequently to LV wall thickness and chamber volume [39]. The model has been validated by reproducing clinical trials in LV hypertrophy [39] and HFrEF patients [44–46].

In this study, we utilized this model to investigate mechanisms of HFpEF. We first evaluated the relative contributions of the commonly postulated HFpEF mechanisms in producing a resting state of HFpEF. We then investigated how these mechanisms alter mechanical stress

and strain in the myocardium, and how these mechanical signals may produce the distinctly different cardiac remodeling patterns in HFpEF versus HFrEF, even while cardiac filling pressure is similarly elevated in both. Thirdly, we evaluated the role of titin stiffening, compared to changes in collagen type and content, on mechanical stress and strain felt by the myocardium in HFpEF. Mathematically understanding the mechanisms of HFpEF is a step toward better understanding the disparate response to therapies between HFpEF and HFrEF, and may improve our ability to design therapies for HFpEF in future.

## Methods

### Mathematical model

We utilized a previously published cardiorenal model, summarized schematically in Fig 1, that integrates a cardiac-ventricular function model [42,43], originally developed by Bovendeerd et al. [47] and Cox et al. [48] (Fig 1A and 1C) and a model of renal function and volume [49–51] (Fig 1D–1G). Briefly, the cardiac portion of the model describes the dynamics of the cardiac cycle (Fig 1A), adaptation of myocytes and remodeling of the left ventricle in response to changes in mechanical loading (Fig 1B), and cardiovascular hemodynamics (Fig 1C). The renal and volume homeostasis portion of the model describes blood flow and pressure through the renal vasculature (Fig 1D); renal filtration, reabsorption, and excretion of sodium, water, and glucose (Fig 1F); whole-body fluid/electrolyte distribution (Fig 1E); and key neurohormonal and intrinsic feedback mechanisms (Fig 1G). The cardiac and renal components of the model are coupled through mean arterial pressure (MAP), venous pressure, and blood volume (BV). MAP and venous pressure calculated from the systemic circulation are inputs to the kidney and are determinants of renal blood flow and glomerular hydrostatic pressure in the kidney model (Fig 1D). The renal model determines sodium and water excretion (Fig 1F), which then determines blood and interstitial fluid volume (Fig 1E), and blood volume feeds back into the circulation model (Fig 1C). Model equations, parameters and initial conditions have been described in detail previously [39] and are given in the supporting material, S1 Text and S1–S6 Tables.

### Mathematical definition of HFpEF

In order to evaluate mechanisms that contribute to HFpEF, a minimal set of criteria were specified to define HFpEF mathematically. By definition, HFpEF requires ejection fraction greater than 50%. In addition, HFpEF is differentiated from other conditions by the presence of elevated cardiac filling pressures [10,52,53]. Clinical diagnostic guidelines for HFpEF [54] recommend using diagnostic score that takes into account multiple echocardiographic measurements and natriuretic peptide levels (NPs) (after first ruling other causes). The key components of this score - elevated E/e', increased left atrial volume index (LAVI), and elevated N-terminal pro-brain natriuretic peptide (NT-proBNP) or BNP - are all surrogate indicators of elevated cardiac filling pressure. If this score is inconclusive, invasive measures of resting LV EDP or pulmonary capillary wedge pressure (PCWP) are used to confirm diagnosis. Thus, elevated filling pressures are a critical defining feature of HFpEF. While some HFpEF patients may have reduced cardiac output, CO is often normal [55], especially at rest, and is not a key component of the diagnostic criteria [54].

Thus, the minimal criteria for HFpEF were considered as: 1) left ventricle end diastolic pressure (LVEDP) $\geq$ 20 mmHg and 2) EF$\geq$ 50%. For this analysis, we focused on elevation of LVEDP at rest. In the early stages of HFpEF, filling pressures may be normal at rest and may only become elevated with exertion [9], but this group has much better (near age-normal) prognoses compared to patients with elevated filling pressures [56], and we did not consider them in this analysis.

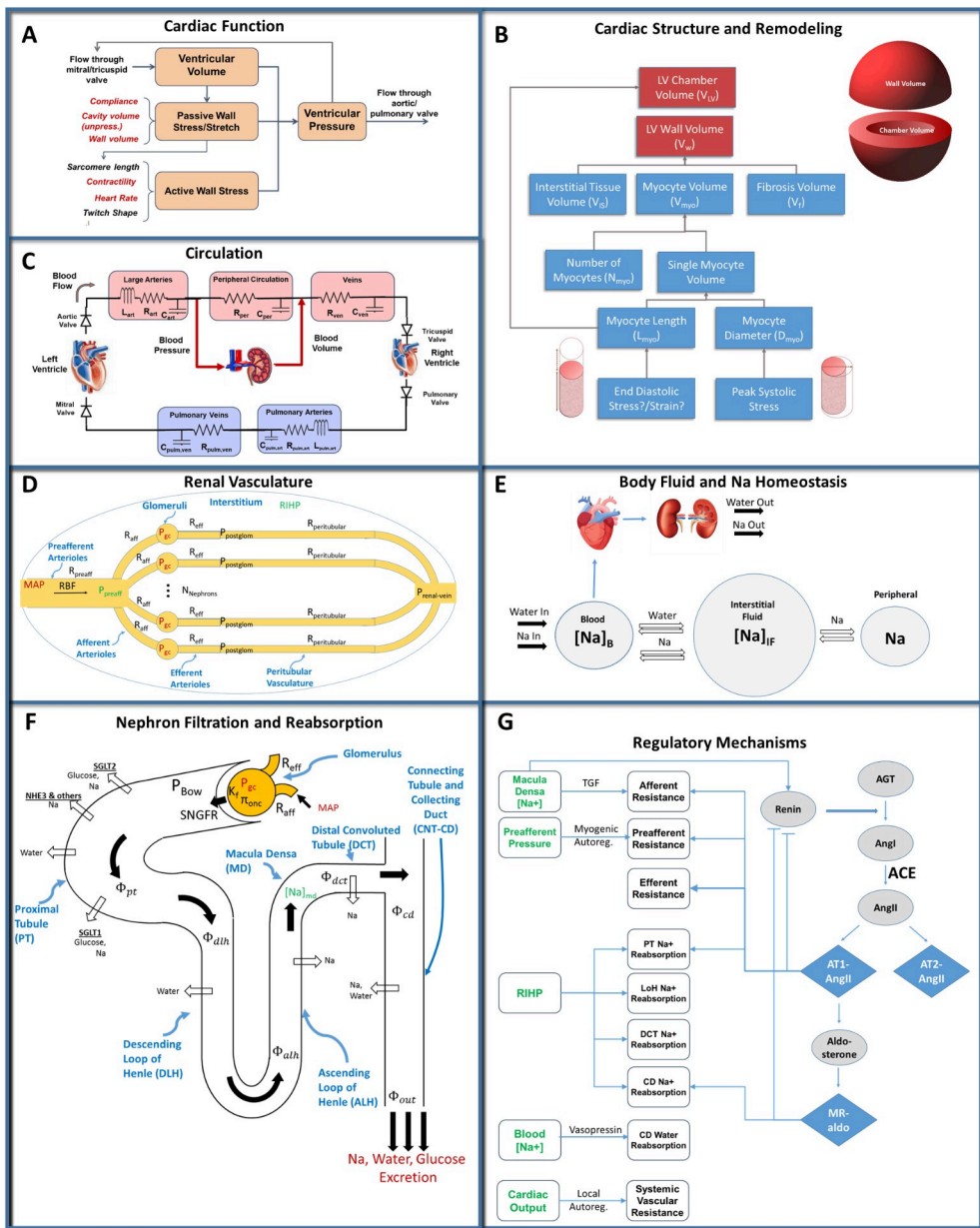

**Fig 1. Schematic of the integrated cardiorenal model.** Model links cardiac mechanics (A) and ventricle remodeling (B), a lumped parameter description of cardiovascular circulation (C), whole body Na⁺ and fluid homeostasis (E), renal hemodynamics (D), renal filtration and reabsorption (F), and neurohormonal and intrinsic feedbacks including the renin-angiotensin-aldosterone system (RAAS) (G). Adapted from Yu et al. (2020) [42].

## Modeling potential mechanisms of HFpEF

To determine the relative contributions of proposed mechanisms to producing a state of HFpEF, the effect of each potential pathophysiological mechanism on LVEDP and EF was evaluated by changing model parameters associated with that mechanism, as summarized in Table 1 and described below.

**Table 1. Parameters varied to simulate pathophysiological mechanisms of HFpEF.**

| Pathophysiological Mechanism | Parameter | Definition | Normal Value | Units C | Simulated Change | Eq. |
|---|---|---|---|---|---|---|
| *Cardiac Changes* | | | | | | |
| Increased myocardial stiffness | $c_f$ | LV fiber stiffness | 11 | - | +0–75% | 1 |
| Impaired contractility | C | Intrinsic LV contractility | 1 | –- | - 0–20% | 3 |
| Outward remodeling of the left ventricle wall | $\Delta L$ | Myocyte length increase from normal | 0 | μm | + 0–25% of baseline L | 6,7 |
| Impaired ventricular relaxation during early diastole | $\Delta t_f$ | Increase in diastolic relaxation time | 0 | seconds | +0–325% of normal $t_{twitch}$ | 9 |
| *Vascular Changes* | | | | | | |
| Vascular Stiffening | $C_{art}$ | Arterial *compliance* | 1.5 | ml/mmHg | -0–50% | 10 |
| | $C_{ven}$ | Venous compliance | 26.7 | ml/mmHg | -0–50% | 10 |
| | $C_{art,pulm}$ | Pulmonary arterial compliance | 2.7 | ml/mmHg | – 0–50% | 10 |
| | $C_{ven,pulm}$ | Pulmonary venous compliance | 22 | ml/mmHg | -0–50% | 10 |
| Reduced venous capacitance | $V_{ven,0}$ | Venous capacitance | 3.0 | L | -0–50% | 10 |
| Hypertension | $R_{preaff}$ | Preafferent arteriole resistance | 14 | mmHg-min/L | + 0–80% | S1 |
| | $R_{aff}$ | Afferent arteriole resistance | 10 | mmHg-min/L | +0–80% | S1, S2 |
| | $R_{eff}$ | Efferent arteriole resistance | 51 | mmHg-min/L | +0–80% | S1, S2 |
| | $K_f$ | Glomerular ultrafiltration coefficient | 4.0 | ml/min/mmHg | -0–40% | S5 |
| | $\eta_{pt}$ | Fractional PT sodium reabsorption | 66 | % | +0–8% | S14 |
| | $\eta_{cd}$ | Fractional CD sodium reabsorption | 84 | % | +0–8% | S16 |
| | RIHP0 | Setpoint for renal interstitial hydrostatic pressure control of sodium reabsorption | 9.7 | mmHg | +0–8% | S47 |

## Increased LV passive stiffness

As described previously, the myocardium was modeled as a homogenous orthotropic material, defined by a fiber stiffness and radial stiffness parameter [42,47]. The stress along the fiber $\sigma_f$ is a nonlinear function of the fiber stretch:

$$\sigma_f = \beta * \left( e^{c_f * (\lambda_f - 1)} - 1 \right) \tag{1}$$

where $\beta$ is a scaling constant, $c_f$ is the stiffness constant along the fiber direction, $\lambda_f$ is the myocardial fiber stretch. As described previously [47], myocardial fiber stretch is related to dynamic LV volume by:

$$\lambda_f = \left( \frac{V_{lv} + \frac{1}{3} V_w}{V_{lv,\ zero\ -p} + \frac{1}{3} V_w} \right)^{\frac{1}{3}} \tag{2}$$

where $V_{lv}$ and $V_{lv,zero-p}$ are LV chamber volumes (i.e. volume of blood in the chamber) under pressurized and zero-pressure conditions respectively, and $V_w$ is the LV wall volume (i.e. volume of LV myocardial tissue, which is the sum of myocyte volume and extracellular matrix volume [ECM]). In the first part of this analysis (sensitivity analysis), the effect of increased myocardial stiffness was evaluated by increasing $c_f$, without differentiating between myocardial and ECM stiffness. Later, we revisit the homogenous assumption and update the model to

distinguish the contributions of myocyte stiffness, ECM stiffness, and their relative volume fractions.

## LV contractility

Contractility $c$ is defined as an intrinsic property of the myocardium. This parameter was decreased to represent reduced myocyte contractile ability (Table 1). As developed originally by Bovendeerd et al [47], contractility is used to calculate active fiber stress in the ventricle wall during systole:

$$\sigma_{active}(c, l_s, v_s, t_a) = c\sigma_{ar} f(l_s)g(t_a)h(v_s) \tag{3}$$

where c is chamber contractility; $\sigma_{ar}$ is a scaling constant; $f(l_s)$ is a sigmoidal function of the sarcomere length $l_s$; $g(t_a)$ is a sinusoidal signal describing cardiac excitation as a function of time elapsed since activation $t_a$; and $h(v_s)$ is sarcomere fiber shortening velocity $v_s$. All equations for determining sarcomere length, time elapsed since cardiac excitation, and shortening velocity have been described previously [39,47] and are given in the S1 Text.

## LV outward dilatation

While concentric remodeling rather than outward dilatation is more often observed in HFpEF patients, some patients may have some degree of outward dilatation. In addition, understanding differences in outward dilatation may be important in distinguishing between HFpEF and HFrEF mechanisms. Thus, we also sought to evaluate the effect of outward dilatation on EF and LVEDP. As described previously [39], outward dilatation is represented in the model as elongation of myocytes resulting in enlargement of the LV cavity volume $V_{lv,zero-p}$.

Myocytes are modeled as cylinders, and myocyte volume is given by:

$$V_{myo} = N_{myo}\left(\frac{\pi D_{myo}^2 L_{myo}}{4}\right) \tag{4}$$

where $N_{myo}, D_{myo}, L_{myo}$ are the number, average diameter, and average length of myocytes, respectively. $D_{myo}$ is the sum of the normal (healthy) diameter $D_{myo,0}$ and any change in diameter $\Delta D_{myo}$ resulting from remodeling by addition of sarcomeres in parallel. $L_{myo}$ is the sum of normal (healthy) length $L_{myo,0}$ and any change in length $\Delta L_{myo}$ resulting from remodeling by addition of sarcomeres in series.

$$D_{myo} = D_{myo,0} + \Delta D_{myo} \tag{5}$$

$$L_{myo} = L_{myo,0} + \Delta L_{myo} \tag{6}$$

As the myocytes elongate, the cavity volume increases:

$$V_{lv,\ zero-p} = V_{lv,0} * \left(1 + \frac{\Delta L_{myo}}{L_{myo,0}}\right)^3 \tag{7}$$

$V_{lv,0}$ is the unpressurized LV cavity volume under normal conditions, before any remodeling. The effect of outward dilatation was evaluated by increasing the change in myocyte length, $\Delta L_{myo}$.

 

## Slowed LV relaxation

Cardiac excitation is modeled as a sinusoidal function, adapted from the form used in Bovendeerd et al [47] to allow the rise time and fall time of excitation to be specified separately:

$$g(t) = \begin{cases} sin^2\left(\pi \frac{t}{t_{twitch}}\right), & t < t_r \\ cos^2\left(\pi \frac{(t - t_r)}{t_{twitch} + \Delta t_f}\right) & t_r \leq t \leq t_{twitch} + \frac{\Delta t_f}{2} \\ 0 & t_{twitch} + \frac{\Delta t_f}{2} \leq t \leq t_{beat} \end{cases} \quad (8)$$

where t is elapsed time since the beginning of activation, $t_{twitch}$ is the duration of LV contraction, $t_r$ is the duration of the rising part of the contraction, and $t_{beat}$ is the duration of a single heart beat (1/heart rate). To model slowed relaxation, twitch time is defined as the sum of the rise time $t_r$ and normal fall time $t_f$:

$$t_{twitch} = t_r + t_f \quad (9)$$

$\Delta t_f$ is the incremental increase in the fall time. When $\Delta t_f$ is zero, the rise and fall is symmetric, and this expression simplifies to the same expression used in [47]. Increasing $\Delta t_f$ causes the excitation signal g(t) to fall more slowly, as illustrated in Fig 2.

Isovolumic relaxation time (IVRT) is normally around 80 ms, and the relaxation time constant τ is typically 35–48 ms, but both are often increased in pressure-overload subjects and in most but not all HFpEF subjects, with IVRT values ranging from 85ms-160ms [10,25] and τ ranging from 45–90 ms [19]. In the model, increases in IVRT and τ (which can be calculated from simulated LV pressure waveforms) can be induced by increasing the parameter $\Delta t_f$. Thus, to evaluate the effect of slowed relaxation, $\Delta t_f$ was increased over a wide range, ranging from 0 to 130 ms, which corresponds to a $t_{twitch}$ of 40 to 170 ms (0–325% increase). This range produced increases in IVRT and τ ranging from their baseline values of 80 ms and 35 ms, respectively, up to extremely large values of 432 ms and 220 ms, respectively. Note that $t_{twitch}$ represents the total time during which the ventricle is contracting and relaxing, while the outputs IVRT and τ represent calculated features of a portion of the LV pressure waveform, and thus do not scale linearly with $t_{twitch}$.

## Systemic vascular stiffening

For each segment *i* of the vasculature, the compliance C defines the relationship between pressure and volume:

$$V_i = V_{i,0} + C_i * P_i \quad (10)$$

where $V_i$ and $P_i$ are the dynamic volume and pressure in each segment, and $V_{i,0}$ is the volume under zero-pressure conditions. Vascular stiffening was modeled as a decrease in compliance C, and was evaluated for four vascular segments i: systemic arterial, systemic venous, pulmonary arterial, and pulmonary venous.

## Reduced venous capacitance

Reduced venous capacitance was modeled as a decrease in the venous unpressurized volume $V_{ven,0}$ in Eq 10 above.

 

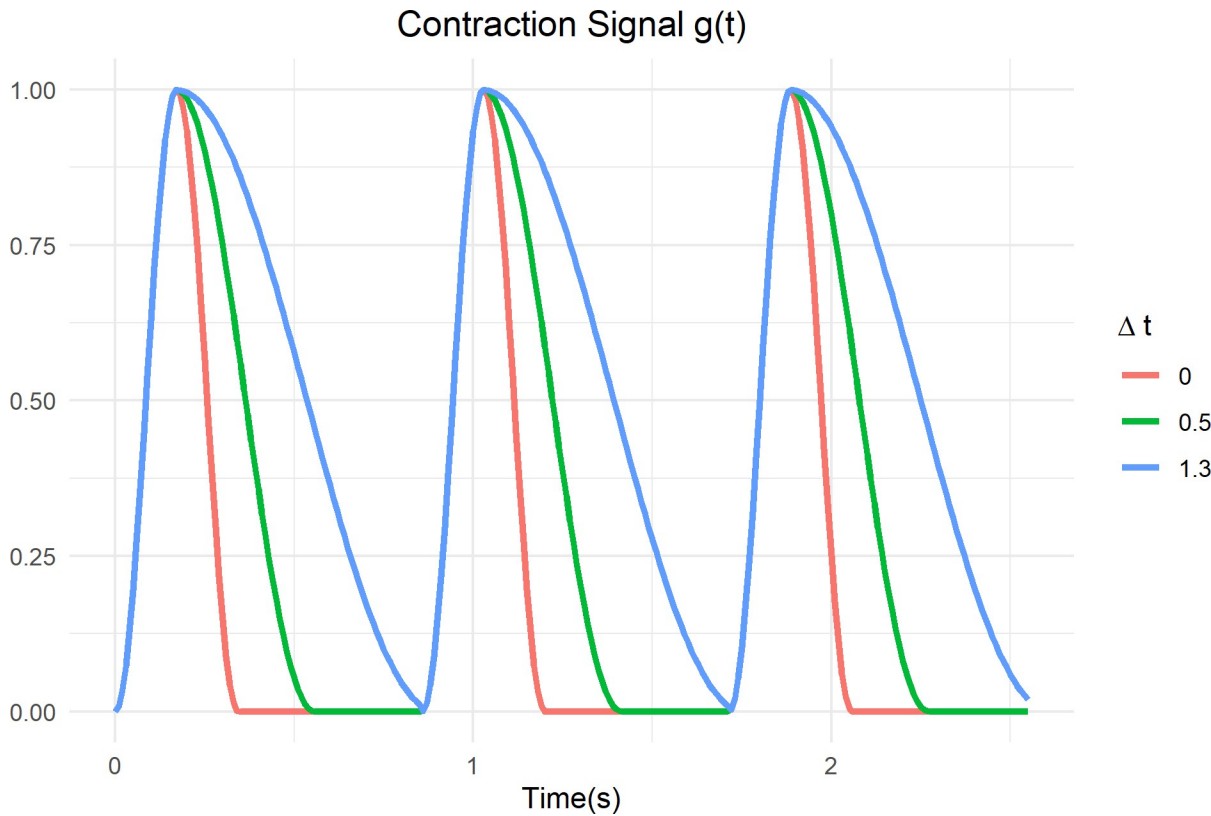

**Fig 2. Slowed relaxation modeled as an elongation of the falling arm of the contraction signal.**

### Hypertension

As described previously [43], hypertension was induced mathematically by changing renal functional parameters that result in sodium and water retention: increasing renal vascular resistance, decreasing glomerular permeability, increasing tubular sodium reabsorption rates, and increasing the set-point for renal interstitial hydrostatic pressure (RIHP) that controls the kidney's pressure-natriuresis response. Each of these mechanisms alone cause small increases in MAP, but when combined, can increase blood pressure over a wide range. Hypertension is a heterogeneous disease, and the mechanisms used here are one way to induce hypertension, but are not intended to represent all forms of hypertension that may exist in HFpEF patients.

### Sensitivity analysis

To evaluate the contributions of each mechanism in producing a state of HFpEF, a sobol global sensitivity analysis was conducted. The sobol method decomposes the variance of a given model output into the fractions attributable to each input. The first order sobol indices quantify the fraction of the variance attributable to each parameter alone, and the total order indices quantify the fraction of the variance due to each parameter jointly with other parameters. The parameters in Table 1 were used as inputs, and LVEDP and EF were considered as model outputs. For each parameter combination, the model was simulated for 60 days (about 1 minute computation time–see S2 Text)–a duration sufficient to allow the model to settle to a new stable state. During this time, the process of outward remodeling of the myocardium over time in response to elevated wall stress (see Eq 11 later) were turned off by setting its rate constant $K_l$

to zero. The sobol analysis was conducted using the sensobol package in R [57]. An initial sampling of $N = 2^9$ was used, and the total number of parameter sets tested was $2N*(1+p)$ or 9216 for $p = 8$ parameters. Sobol indices were estimated using the Monte-Carlo estimator of Azzini et al [58], and confidence intervals were determined by bootstrapping with 1000 replicates.

## Simulating HFpEF over time: Evaluating mechanical drivers of cardiac remodeling

After evaluating combinations of mechanisms that can produce a state of HFpEF (elevated LVEDP with normal EF), we next investigated mechanisms of LV remodeling over time that may differentiate HFpEF from HFrEF. Specifically, outward dilatation is generally thought to be a response to increased preload. We sought to understand why, once preload becomes elevated in HFpEF patients, it does not cause outward dilatation, chamber enlargement, and progressive decline in EF that is seen in HFrEF.

Cardiac remodeling occurs as a result of cellular and tissue adaptations to different loading conditions. Although the different cellular or tissue adaptations for cardiac growth and remodeling have been well studied [59], the specific mechanical stimuli that evoke a remodeling response is still debated [60]. It has long been recognized, based on work by Grossman et al [61], that ventricle wall thickening appears to normalize peak systolic wall stress during systole, while outward dilatation appears to occur in response to elevated passive wall stress during diastole [61,62]. Grossman proposed that the myocardium remodels in response to changes in peak systolic stress $\sigma_{f,peak}$ by adding or removing sarcomeres in parallel, thus increasing myocyte diameter and wall thickness, while it remodels in response to an increase in passive diastolic stress $\sigma_{f,ED}$ by adding sarcomeres in series, increasing myocyte length and causing outward dilatation [61]. We previously implemented this hypothesis by allowing myocyte diameter and length to change over time in response to LV wall stress (Fig 1B) and demonstrated that with these two simple remodeling laws, the model could reproduce the distinct remodeling responses to pressure and volume overload [39], including the progressive outward remodeling in HFrEF [42] and the regression of remodeling with anti-hypertensive therapies in LV hypertrophy and HFrEF patients [43,44].

However, more recent studies propose that diastolic wall stretch is the actual stimulus for outward dilatation [63–66]. Here, we revisited the assumptions regarding the driver of myocyte lengthening. Specifically, we alternatively considered passive wall stress and passive stretch as the possible signals for outward dilatation. Using two different versions of the model described below, we evaluated the remodeling response over time to reduced LV contractility (the mechanism most commonly associated with HFrEF) versus increased LV stiffness (the mechanism most commonly associated with HFpEF).

## Model 1: Stress-driven remodeling

In this version of the model, as we have described previously [39,61], myocyte length increases when end diastolic longitudinal wall stress $\sigma_{f,ED}$ exceeds the upper limit of normal $\sigma_{f,ED,0}$. Increases in myocyte length were approximated as irreversible (i.e. the change in myocyte length does not return to normal if $\sigma_{f,ED}$ is returned to normal levels), since previous work has suggested that regression of myocyte elongation may occur much more slowly than progression [39,42].

$$\frac{d\left(\Delta L_{myo}\right)}{dt} = max\left(K_l\left(\frac{\sigma_{f,ED}}{\sigma_{f,ED,0}} - 1\right), 0\right) \qquad (11)$$

Here $\Delta L_{myo}$ is the change from baseline in average myocyte length and $K_l$ is the rate constant governing the speed of elongation.

## Model 2: Stretch-driven remodeling

In this version of the model, myocyte length increases when end diastolic wall longitudinal stretch $\lambda_{f,ED}$ exceeds the upper limit of normal $\lambda_{f,ED,0}$.

$$\frac{d\left(\Delta L_{myo}\right)}{dt} = max\left( K_l\left(\frac{\lambda_{f,ED}}{\lambda_{f,ED,0}} - 1\right), 0\right) \tag{12}$$

Where $\Delta L_{myo}$ is the change from baseline in average myocyte length, and $K_l$ is the rate constant governing the speed of elongation.

## Simulation procedure

Because a history of hypertension is common in both HFpEF and HFrEF, parameters were first changed to induce mild hypertension (see parameters in Table 1). After an equilibration period that allowed the model to reach a stable state, during which outward remodeling was turned off (rate constant $K_l = 0$ in Eqs 11 and 12), LV contractility was decreased by 15% to simulate HFrEF, and myocardial stiffness $c_f$ was increased by 60% to represent HFpEF. Outward remodeling was "turned on" ($K_l$ set to $> 0$) and the simulation was run for 1 year. This simulation was performed with two versions of the model described above, with outward remodeling driven either by LV end diastolic stress (model 1 –Eq 11) or by LV end diastolic stretch (model 2 –Eq 12).

## Evaluating contribution of myocyte and collagen stiffness and volume fraction

To enable the effects of myocyte stiffening, collagen stiffening, or increased volume fraction on the development of HFpEF to be evaluated separately, we updated the model to approximate the myocardium as a layered composite of unidirectional fibers of myocytes and extracellular matrix in parallel, each component defined by a stiffness constant and volume fraction. This is an iso-strain condition, in which a force applied to the composite produces the same strain but different stresses (if stiffness constants are different) in each component material. When the chamber is pressurized, the total force on a segment of myocardial tissue is the sum of the forces on the myocardium and extracellular matrix:

$$F_{total} = F_{myo} + F_{ecm} \tag{13}$$

where $F_{myo}$ and $F_{ecm}$ are the forces on the myocytes and extracellular matrix, respectively. Since $F = \sigma * Area$, the composite passive stress $\sigma_f$ can be shown to be:

$$\sigma_f = \sigma_{f,\,myo} * \frac{A_{myo}}{A_{total}} + \sigma_{f,\,ecm} * \frac{A_{ecm}}{A_{total}} \tag{14}$$

where $A_{myo}/A_{total}$ and $A_{ecm}/A_{total}$ are area fractions for myocytes and ECM respectively, and are the same as the volume fractions of each component.

As a simplifying assumption, the stress-stretch relationship for each material are assumed to follow the same nonlinear relationship:

$$\sigma_{f,myo} = \beta * \left( e^{c_{f,myo} * \left( \lambda_f - 1 \right)} - 1 \right) \tag{15}$$

$$\sigma_{f,ecm} = \beta * \left( e^{c_{f,ecm} * \left( \lambda_f - 1 \right)} - 1 \right) \tag{16}$$

where $\beta$ is a fitting constant, $c_{f,myo}$ and $c_{f,ecm}$ are stiffness constants for myocytes and ECM respectively, and $\lambda_f$ is the LV chamber stretch. Under homogeneous conditions, $c_{f,myo}$ and $c_{f,ecm}$ were assumed to be the same ($c_f$ in Table 1), and values for $C_f$ and $\beta$ were taken from [47].

Thus, increased myocyte stiffness (e.g. due to titin hyperphosphorylation) was modeled by increasing $c_{f,myo}$; increased stiffening of the extracellular matrix (e.g. due to increased collagen crosslinking or deposition of amyloid complexes) was modeled by increasing $c_{f,ecm}$; and increased collagen volume fraction was modeled by increasing $A_{ecm}$ and $A_{total}$ by the same amount.

## Technical implementation

The model was implemented in R v4.1.1 and utilizes the RxODE package [67]. More information on the technical implementation is given in S2 Text.

## Results

### Contribution of potential mechanisms in producing a state of HFpEF

Sobol sensitivity analysis identified LV stiffness, hypertension, and LV contractility as the most important determinants of LVEDP (Fig 3A). Arterial compliance, outward dilatation, venous capacitance, and venous compliance had much weaker relative effect. Slowed LV relaxation was not found to be important by itself (first order indices not different from zero), but had a small interactive effect with other parameters (non-zero total order index).EF was most strongly influenced by outward dilatation (Fig 3B, first order index > 75%). LV contractility, hypertension, and arterial compliance had much weaker effects.

Fig 4 further illustrates the effect of changing the parameters identified as most important, alone and in combinations, on LVEDP and EF (A and B). The bottom left corner of each panel in these figures shows the reference state, when all parameters are at their normal value. The bottom panel (C) shows the resulting heart failure state, based on the defined LVEDP and EF criteria. First, a state of HFpEF only occurred when LV stiffness was increased and when outward dilatation was limited. Outward dilatation of even 10% tended to decrease EF below 60%, and larger increases resulted in ejection fractions below 50%, even without changes in other parameters. This is perhaps unsurprising, since outward dilatation represents an increase LV chamber volume, and LV end diastolic volume is the denominator of EF. We will revisit the factors determining outward dilatation later.

When outward dilatation was limited (first two rows of each panel), a large increase in LV stiffness alone (75%) was sufficient to produce a state of HFpEF, but HFpEF could also occur at lower degrees of LV stiffening, when combined with either hypertension, reduced arterial compliance, or reduced LV contractility. However, when large reductions in contractility were combined with large reductions in arterial compliance plus hypertension, ejection fraction tended to decrease into the intermediate range (heart failure with midrange ejection fraction, HF-mEF), even if LV stiffness was normal.

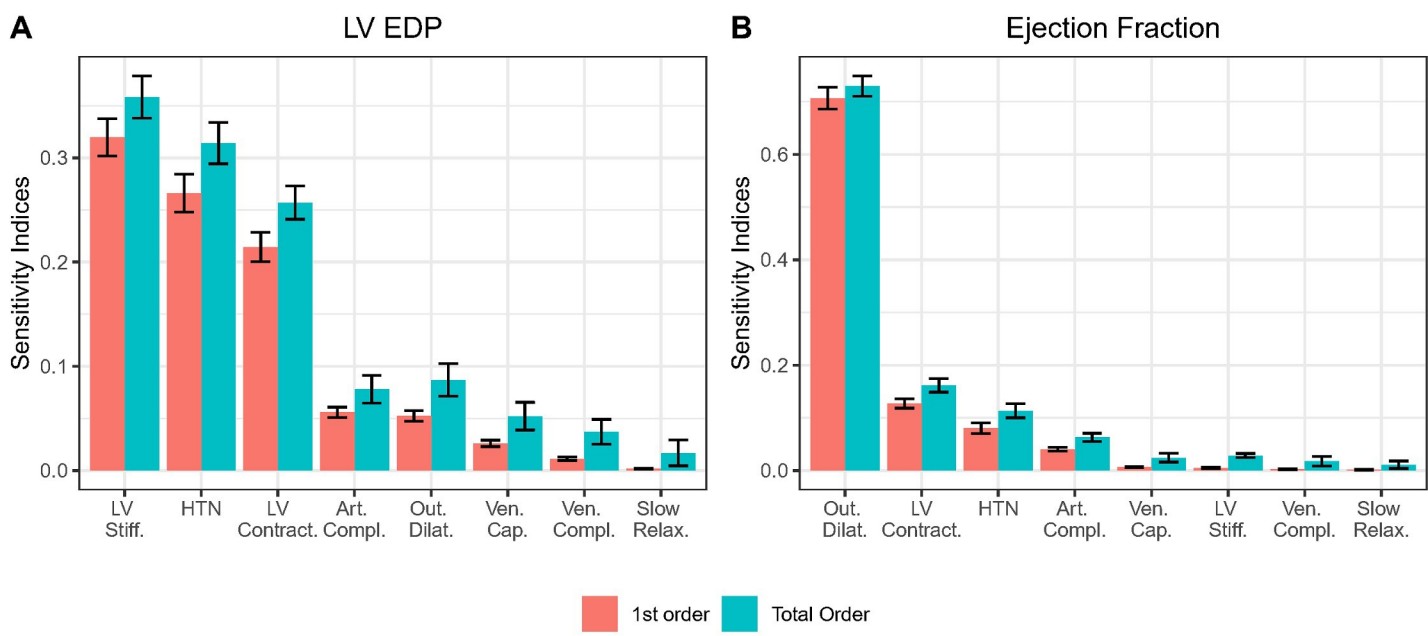

**Fig 3. Sobol sensitivity of (A) LVEDP and (B) LV EF to changes in each potential HFpEF mechanism (Error bars: 95% confidence interval).**

The trends in interstitial fluid volume were very similar to the trends in LVEDP (Fig 5A), and tended to increase as each mechanism worsened, alone and in combination. Cardiac output tended to decrease following the same pattern (Fig 5B).

## Effect of potential mechanisms of HFpEF on LV myocardial stress and strain

The analysis above identified sets of mechanisms that can produce a state of elevated LVEDP without lowering ejection fraction, and indicated that LV stiffness must be at least mildly increased while outward dilatation must be minimal in order to produce a state of HFpEF. In those simulations, the degree of outward dilatation was set to a constant and not allowed to change over time. However, elevated LVEDP necessarily alters the mechanical state felt by the myocardium, with consequences for subsequent ventricular remodeling. Increased preload is generally thought to be associated with outward remodeling and chamber enlargement, yet in HFpEF, preload is elevated while outward dilatation is minimal. Thus, we next sought to understand how potential HFpEF mechanisms may affect mechanical signals for remodeling differently than HFrEF mechanisms, potentially resulting in different remodeling patterns over time.

The two signals commonly proposed to drive outward dilatation are diastolic LV wall stress and strain. As shown in Fig 6A, LV end diastolic stress followed the same pattern as LVEDP, increasing with increased LV stiffness, reduced LV contractility, reduced arterial compliance, and worsening hypertension. LV end diastolic strain (Fig 6B) also increased with reduced LV contractility, reduced arterial compliance, and worsening hypertension. However, the effect of LV stiffness on strain followed a different pattern. When LV contractility and arterial compliance were normal, increased LV stiffness caused small subtle increases in LV strain. But interestingly, when combined with reduced contractility and/or arterial compliance, increased LV stiffness mitigated the rise in LV strain, helping to keep strain at near normal levels.

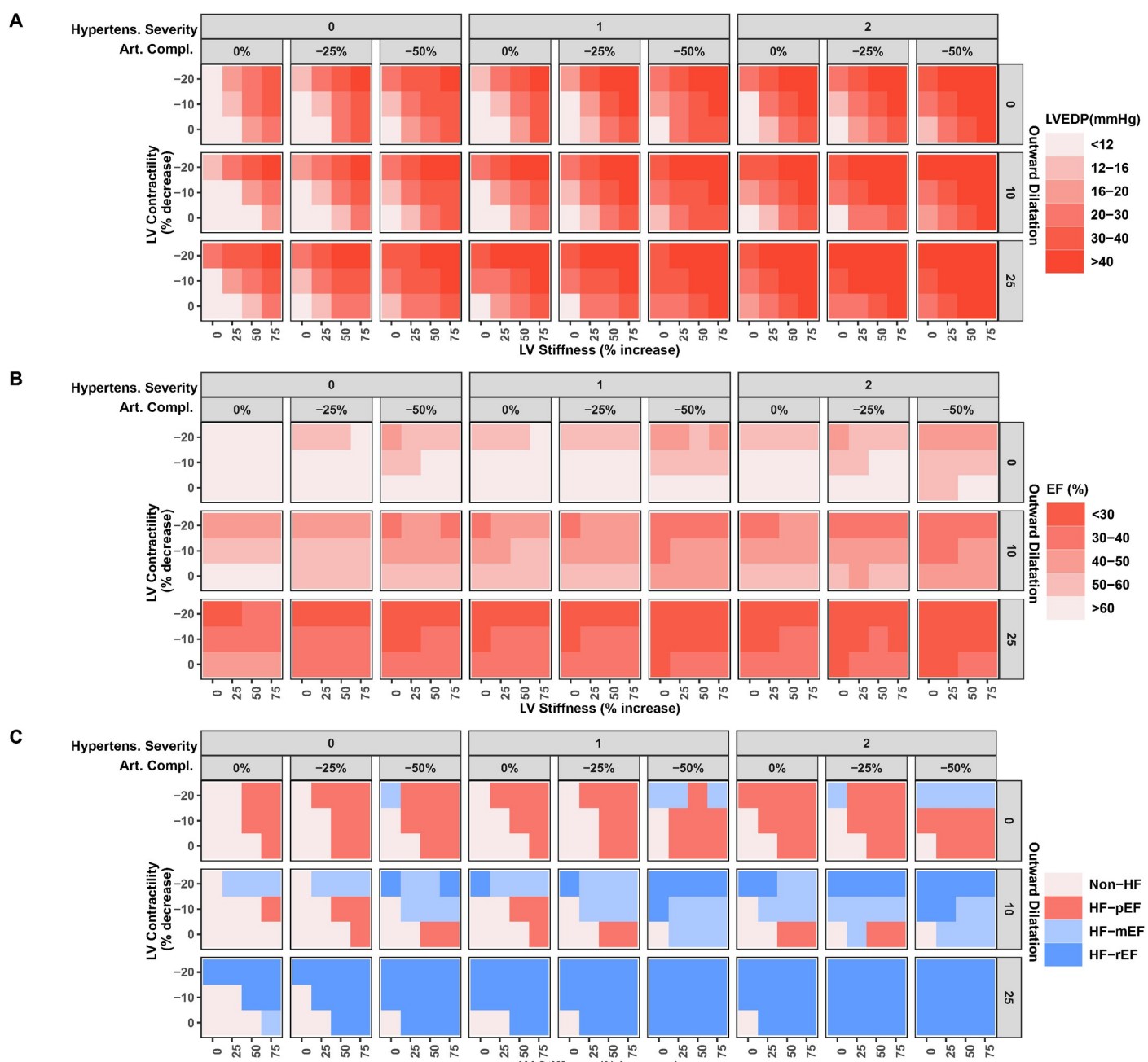

**Fig 4. Effect of changing LV stiffness, LV contractility, hypertension, arterial compliance, and outward dilatation, alone and in combination LV EDP and EF.** A) LVEDP, B) EF. C) HF state produced by each region of the parameter space: *HFpEF* (EF>50% and *LVEDP* > 20mmHg), HF-mEF (40%<EF<50% and *LVEDP* > 20mmHg), HFrEF (EF<40% and *LVEDP* > 20mmHg), and non-HF (*LVEDP* < 20 mmHg).

## LV outward remodeling in response to LV end diastolic stress or strain over time

Given that increased LV stiffness has opposite effects on LV passive stress versus strain, and given that minimal outward dilatation is required to maintain a state of HFpEF, we next investigated the consequences of assuming LV passive stress versus LV passive strain as the signal

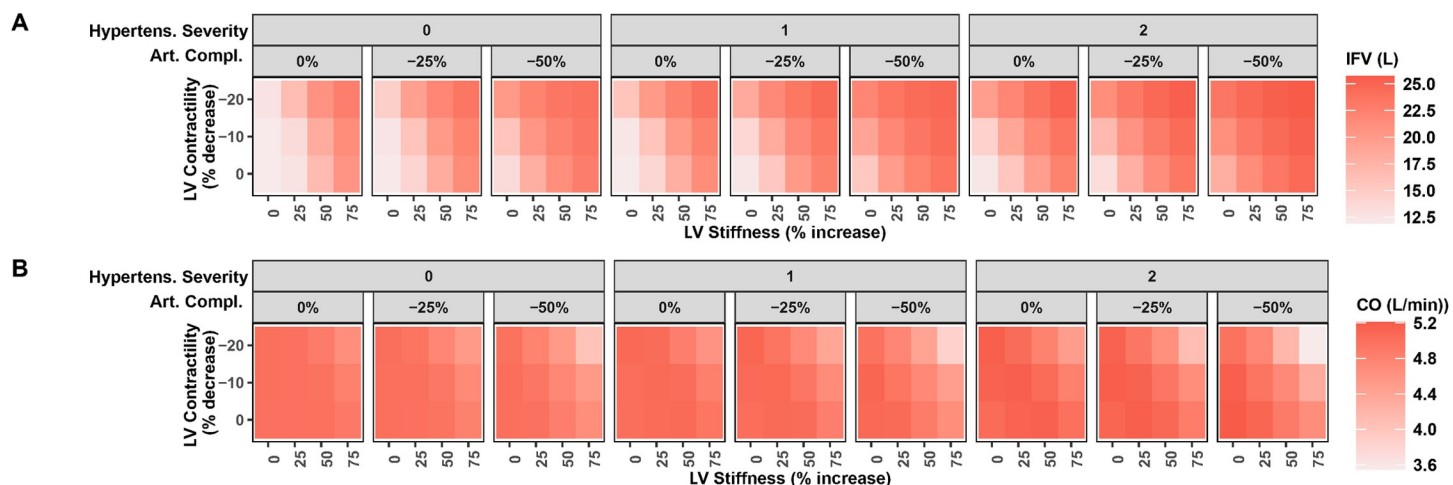

**Fig 5. Effect of changing LV stiffness, LV contractility, hypertension, and arterial compliance, alone and in combination, on A) interstitial fluid volume (IFV) and B) cardiac output (CO).** For these simulations, outward dilatation was zero.

that drives outward dilatation, and whether this could explain the different remodeling patterns found in HFpEF versus HFrEF.

As shown in Fig 7, the effect of either increased LV stiffness (the most common abnormality in HFpEF) or reduced contractility (representing impaired systolic function, the most common abnormality in HFrEF) on LV remodeling over time was evaluated under two different model assumptions: outward remodeling driven either by LV end diastolic stress (model 1 – Eq 11) or LV end diastolic strain (model 2 –Eq 12). For both, initial outward dilatation was assumed zero.

When remodeling was driven by stress, both increased LV stiffness and decreased LV contractility caused EF to progress steadily downward into the HFrEF range. On the other hand, when remodeling was driven by strain, EF still progressed toward HFrEF in response to decreased contractility, but remained stable above 50% (in HFpEF range) in response to

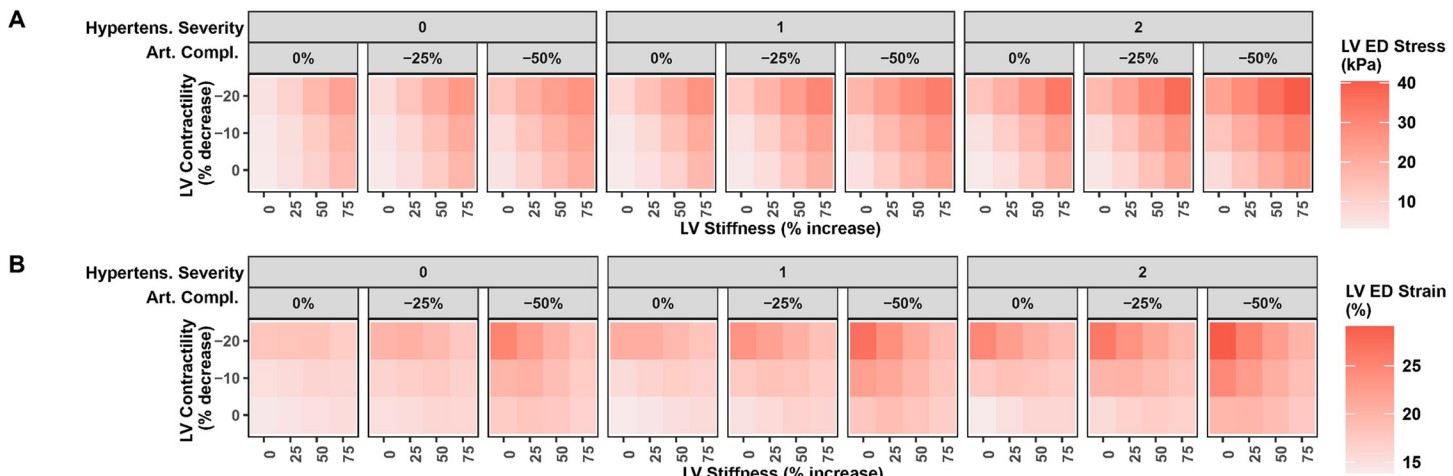

**Fig 6. Effect of HFpEF mechanisms on changes in LV end diastolic stress (A) vs. strain (B).** Reduced contractility, increased outward dilatation, and reduced arterial compliance caused both stress and strain to increase. However, increased LV stiffness caused them to change in opposite directions–stress increased while strain decreased.

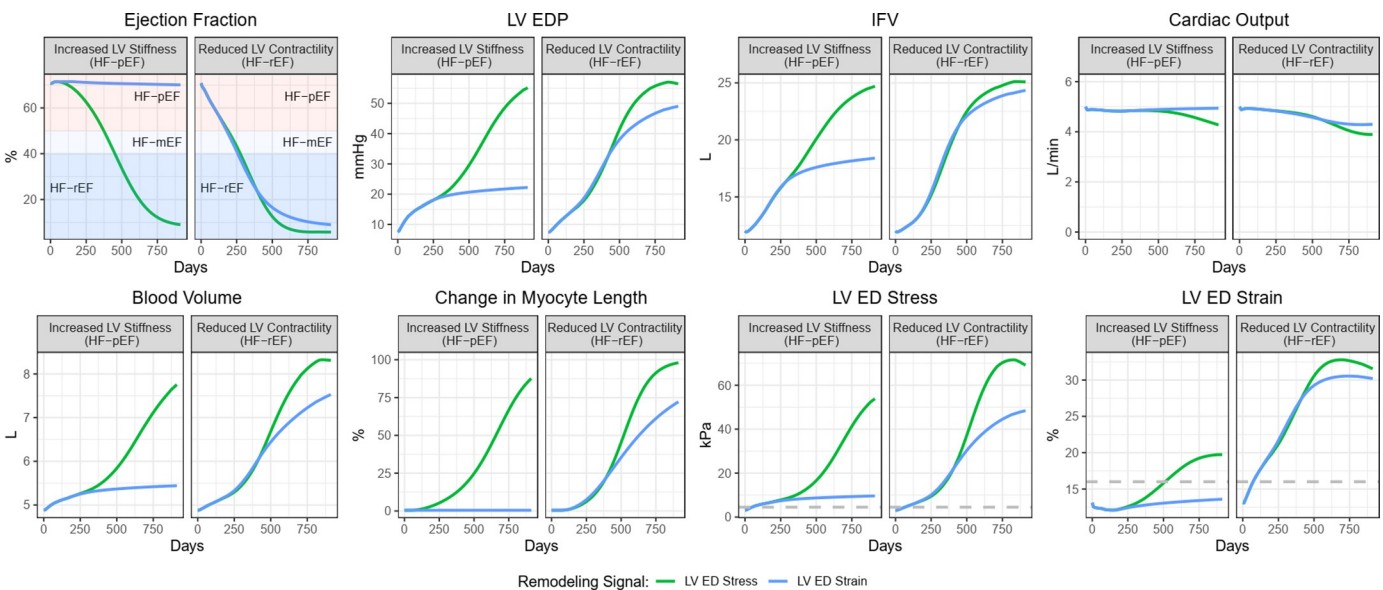

**Fig 7. Effect of increased LV stiffness or reduced contractility (representative of HFpEF and HFrEF, respectively) over time under two different models of remodeling–strain-driven and stress-driven.** With both models, reduced contractility caused EF to decline into the HFrEF range. However, only the strain-driven model allowed EF to remain in the normal range over time with increased LV stiffness. Gray dashed lines are threshold levels of stress/strain above which outward remodeling occurs.

increased LV stiffness. The difference in the EF trajectories between the stress and strain models following increased LV stiffness were a consequence of different degrees of myocyte elongation, which were in turn due to differences in the underlying stress or strain driver (Fig 7 bottom row).

Taken together, the results of the strain-driven model are consistent with the different cardiac remodeling patterns in response to increased LV stiffness (HFpEF) vs. decreased contractility (HFrEF), while the stress-driven model cannot explain the maintenance of normal EF in HFpEF.

## Effect of increased LV myocyte stiffness, ECM stiffness, and collagen volume fraction on cardiac hemodynamics in strain-driven remodeling

Given the importance of LV stiffness, we next sought to understand the impact of changes in intracellular vs extracellular components of the myocardium. While we initially modeled the myocardium as a homogeneous material with a single material stiffness, it is better modeled as a composite material of myocytes and ECM (Eqs 13–16). Myocardial stiffness has been found to increase as much as 2-fold in HFpEF. In addition, insoluble collagen, which is stiffer than soluble collagen, as well as collagen volume fraction, are increased in HFpEF [20].

Fig 8 shows the effect of independently increasing myocyte stiffness (representing titin hyperphosphorylation), collagen stiffness (representing increased collagen cross-linking), and collagen volume fraction, alone and in combination, on LVEDP, LV passive strain, and LV passive stress felt in the myocytes and ECM. For comparison, the effects are shown when systolic function is normal (normal contractility–bottom row in each panel) and impaired (reduced contractility–top row in each panel). While parameters were varied independently, it should be noted that in reality changes in collagen or myocyte stiffness could also impact contractility.

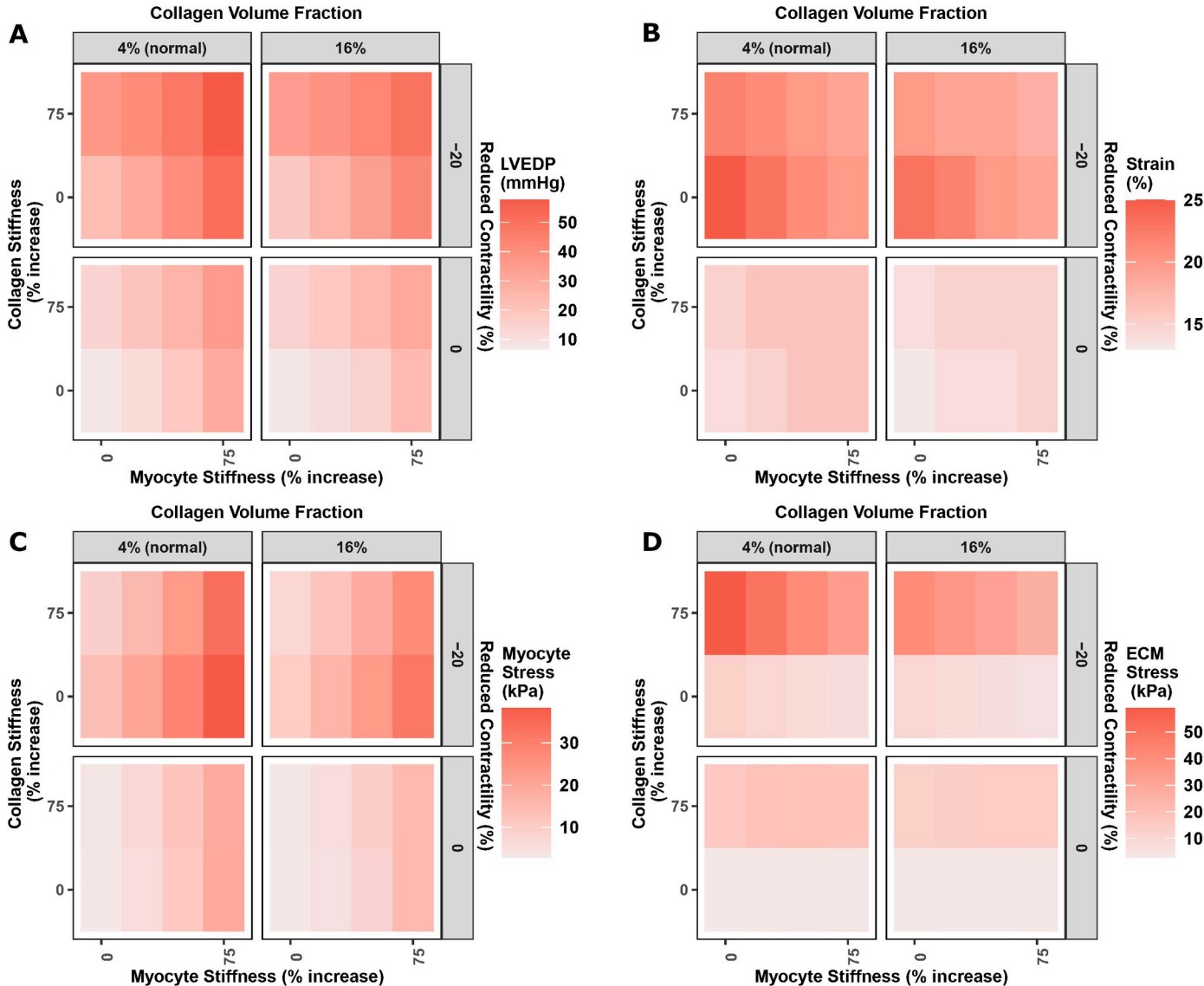

**Fig 8. Effects of collage stiffness, myocyte stiffness, collagen volume fraction, and LV contractility on LV mechanical state.**

Consider first the case in which systolic function is normal (bottom row in each panel) and collagen volume fraction is normal (left column in each panel). As myocyte stiffness increases, LVEDP (A) and myocytes stress (C) go up, while LV strain (B) increases but to a lesser degree. If collagen stiffness also increases, LVEDP is raised further, but there is minimal effect on strain or myocyte stress, as the additional load from the higher pressure is borne by the ECM (D) rather than the myocytes. If collagen volume fraction is increased, there is a very weak reduction in all four measures.

When systolic function is impaired while all else is unchanged (top row, bottom left corner in each panel), LVEDP is somewhat elevated, but the increase in LV strain is much larger–to a degree that would likely be untenable for myocytes. However, if myocyte stiffness is also increased, the rise in strain is much smaller (although the myocyte stress is increased further). And if collagen stiffness is also increased, the rise in both LV strain and myocyte stress are

mitigated, as more of the stress is borne by the ECM. This suggests that while titin stiffening increases LVEDP, it may protect against excess LV strain. It also supports increased collagen stiffness (potentially as a result of increased crosslinking) as a protective mechanism that further reduces the strain, and unloads the myocytes as well.

In addition, if LV strain is the driver of outward remodeling, then if LV titan and/or collagen stiffness are increased *prior to* ischemic or other injury that impairs systolic function, it may allow LV strain to remain low enough to prevent outward remodeling and the decline in ejection fraction observed in HFrEF, but not preventing elevation of LVEDP, thus resulting in a state of HFpEF. In other words, preexisting LV stiffness may protect again outward dilatation and reduced ejection fraction, but not against elevated filling pressures.

## Discussion

HFpEF remains a challenging condition to manage and treat, in part because its underlying mechanisms are heterogenous and not fully understood. Better understanding the role and interaction of the various mechanisms that contribute to HFpEF and differentiate its progression from that of HFrEF may facilitate better treatment and drug development for this patient population. In this study, we utilized a mathematical model of cardiorenal function to quantify the relative contribution of commonly proposed pathophysiologic mechanisms to the development of a state of HFpEF. Our simulations support existing evidence for elevated LV passive stiffness as the critical mechanism contributing to elevated filling pressure in HFpEF, and arterial stiffening (reduced compliance), hypertension, and impaired contractility as exacerbating mechanisms.

We also investigated how these factors affect mechanical signals of cardiac remodeling, and evaluated which combinations of mechanisms and mechanical signals can explain the different remodeling patterns in HFpEF vs. HFrEF over time. This analysis adds to the growing support for LV passive strain, rather than LV passive stress, as the signal for outward dilatation. Further, our simulations suggest that because a stiffer myocardium experiences less strain at a given filling pressure, sustained elevation in filling pressure does not trigger outward remodeling. This means that, when other factors that exacerbate LVEDP, such as impaired contractility, hypertension, or arterial stiffening, occur in an already stiffened heart, outward dilatation may be minimal. Thus, all the signs and symptoms of heart failure resulting from increased filling pressures can occur without a progressive decline in ejection fraction.

### HFpEF as a consequence of LV passive stiffening plus additional insult(s)

The critical role of LV passive stiffening in HFpEF identified here is consistent with the many clinical studies that have shown that passive myocardial stiffness is elevated in HFpEF patients [17–19]. The finding that a severe increase in LV stiffness alone may be sufficient to elevate LVEDP and cause HFpEF is consistent with previous simulations by others with the CircAdapt model [68] (which uses the same foundational cardiac mechanics model [47], but uses fixed CO in a closed system, i.e. no renal control of fluid volume). Here, we further showed that HFpEF can also occur if milder LV stiffening is coupled with another insult: hypertension, arterial stiffening, or a mild reduction in LV contractility.

The deleterious consequences of adding insults such as arterial stiffening or hypertension to a stiffened myocardium are illustrated in Fig 9. The back-up of LV filling pressure into the normally low-pressure venous and capillary beds is responsible for many of the signs and symptoms of heart failure. Increased LV stiffness alone shifts the LV diastolic pressure curve upward (A) and increases stressed blood volume and capillary and venous pressures (C, E, and F). Alone, neither hypertension nor decreased arterial compliance have much effect on LV

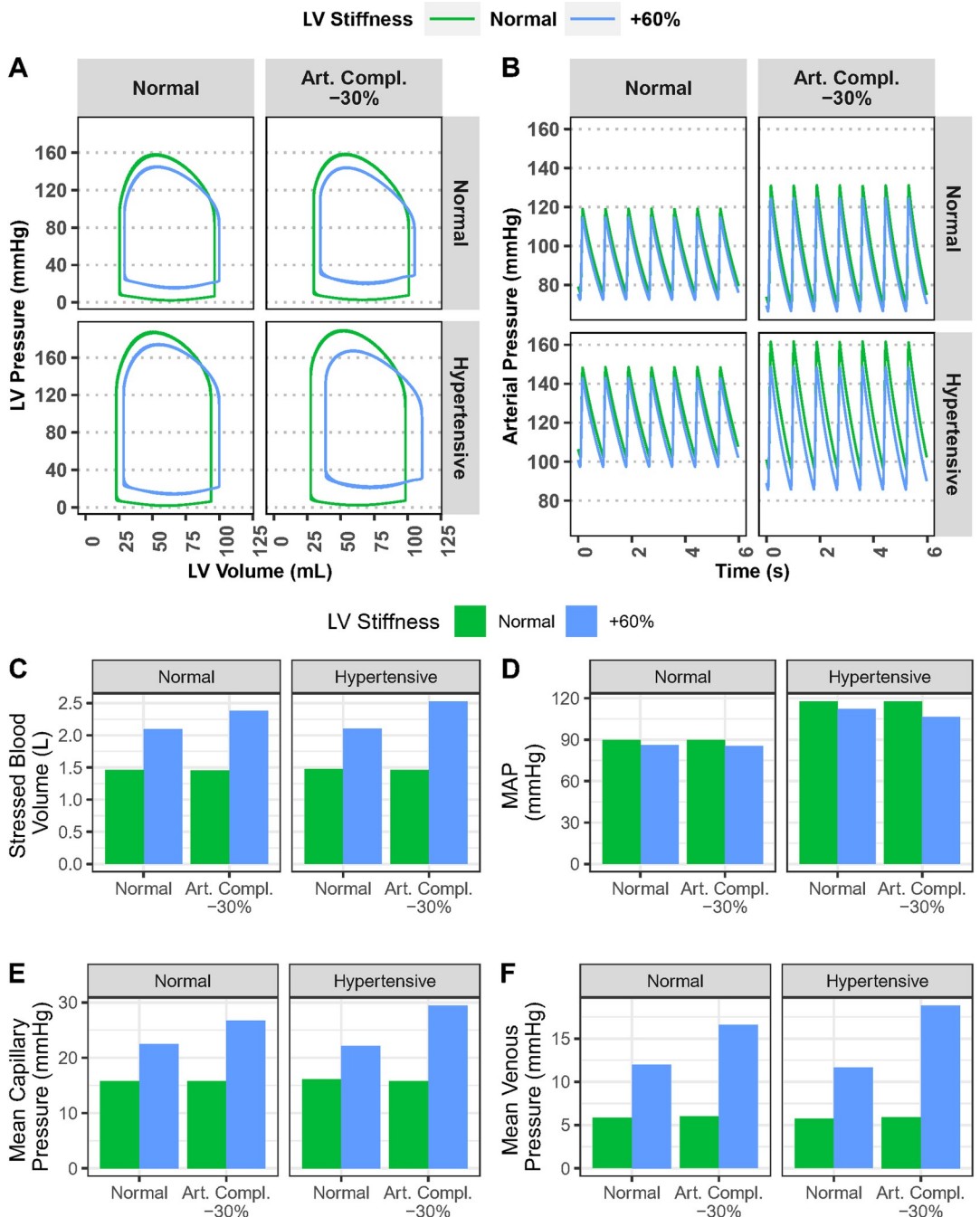

**Fig 9. The deleterious consequences of arterial stiffening and/or hypertension in a stiffened myocardium.** A) LV stiffness shifts the diastolic pressure curve upward; decreased arterial compliance shifts the end systolic pressure upward, thus increasing end systolic elastance ($E_{es}$); hypertension increases both end systolic and peak systolic pressure; Reduced arterial compliance alone increases arterial pulse pressure (B), but has minimal effect on stressed blood volume (C), MAP (D), mean capillary pressure (E), or mean venous pressure (F). However, it has a strong exacerbating effect on stressed blood volume and capillary and venous pressures when combined with a stiff myocardium. Hypertension has a similar exacerbating effect.

diastolic pressure, stressed blood volume, or venous and capillary pressures (although reduced arterial compliance increases arterial pulse pressure (B), while hypertension increases peak systolic pressure and mean arterial pressure (D)). However, when decreased arterial compliance

is added to LV stiffening, the elevations in stressed blood volume and venous and capillary pressures are nearly doubled. And they are further exacerbated if hypertension is also added.

## LV passive stiffening mitigates excessive LV wall strain and explains different remodeling patterns between HFpEF and HFrEF

Our analysis demonstrated that LV passive strain, as opposed to stress, can better explain the different remodeling patterns between HFpEF and HFrEF. While there is still much debate regarding the signals for cardiac tissue growth and remodeling, modeling analyses have increasingly suggested strain as the critical signal for tissue growth during volume-overload [36,60,64]. Most recent analyses use strain or a combination of stress and strain, and our analysis adds further support for strain as the key signal [60,64–66].

These findings support a new hypothesis for the pathophysiology distinguishing HFpEF from both LVH and HFrEF: namely, that LV passive stiffening contributes to a rise in LVEDP but prevents excess LV wall strain (since a stiffer heart stretches less for a given change in pressure), thus limiting outward remodeling in response to other insults to the myocardium and preventing progressive decline in EF. Thus, a state of HFpEF occurs, in which filling pressures are elevated but EF is normal and stable.

This effect can be further understood by considering the mathematical relationships between filling pressure, stress, and strain. For a spherical, strain-stiffening (i.e. stiffness increases with increasing stretch) vessel, passive wall stress $\sigma$ is linearly proportional to filling pressure $P_i$:

$$\sigma \propto P_i \tag{17}$$

while passive wall strain $\varepsilon$ can be shown to be log-linearly proportional to pressure, and inversely proportional to stiffness (see S3 Text for further explanation):

$$\varepsilon \propto \frac{log(P_i)}{c_f} \tag{18}$$

Thus, when filling pressure increases, wall stress will always increase proportionally (Eq 17), and for a given stiffness, strain will also increase. But if stiffness is increased as well, then strain will only increase if the log-increase in pressure is greater than the increase in stiffness (Eq 18).

So, when LV passive stiffness is normal, any factor that increases filling pressures will increase both stress and strain proportionally. [42,44]. On the other hand, when LV passive stiffness is elevated, LVEDP rises (Fig 4) but the rise in LV passive strain is minimal (Fig 6). Any additional insult that elevates filling pressures further (e.g. hypertension, arterial stiffening, reduced contractility) will increase strain less than it would at normal stiffness. If strain is the signal for outward dilatation, then outward dilatation will be much less, and ejection fraction will not decline. The systemic consequences of congestion and elevated filling pressure will still occur, leading to the signs and symptoms of heart failure, but ejection fraction will remain normal.

HF-pEF is associated with multiple co-morbidities (kidney disease, metabolic syndrome, coronary artery disease, atrial fibrillation, valvular disease) [9], and each of these comorbidities may contribute insults such as the ones evaluated here (hypertension, impaired contractility, vascular stiffening, etc.). Our simulations suggest that comorbidities that produce cardiac or vascular changes that are not severe enough to cause HFrEF in a heart of normal stiffness may cause HF-pEF as LV stiffness is increased. Understanding HFpEF mechanisms in this way, as additional insults added to an already stiffened myocardium, provides an important step

forward in the path to understanding and reproducing the heterogeneity within the HFpEF population. We have previously used the same cardiorenal model to simulate some of these comorbidities, including hypertension, chronic kidney disease, diabetes, and mitral/aortic stenosis. Going forward, these comorbidities can be added to a stiffened myocardium to generate virtual HFpEF populations. Indeed, because the model mechanistically couples renal and cardiac function, it is well suited for these types of simulation. In addition, the majority of therapies that are used to manage heart failure act through renal mechanisms. Accounting for the heterogeneity of HFpEF populations is a well-recognized challenge in improving management or treatment of HFpEF, as different patient types may respond differently. Going forward, we can utilize this model both to investigate differential responses to existing therapies (e.g. Why do SGLT2i and MRA improve outcomes while RAAS blockers have failed? Why are these treatments more effective at reducing hospitalizations than mortality? etc.), to understand and predict differential subpopulation responses (Why do some subpopulations see greater benefits than others?), and particularly to suggest new approaches or to predict patient-specific or subpopulation-specific effects of new mechanistic targets under development.

## Limitations and other considerations

This study defined HFpEF as a state of normal ejection fraction and chronically elevated LVEDP. However, some HFpEF patients, especially early on, have impaired exercise intolerance but normal resting filling pressures [9]. These patients tend to have much better prognoses than patients with elevated LVEDP. This study did not explicitly model exercise tolerance. However, both experimental studies [69,70] and previous modeling studies [68] have shown that increased LV stiffness is associated with poorer exercise tolerance.

While slowed LV relaxation is a common finding in HFpEF, the model predicted it does not contribute significantly to elevated LVEDP, consistent with previous simulations by others [68]. Reduced venous capacitance and compliance both had relatively small effects on resting LVEDP alone, suggesting that the system is normally able to accommodate wide variability in venous function, including increases in stressed blood volume, without increasing cardiac filling pressures. However, these mechanisms did influence LVEDP sensitivity to other parameters, suggesting that they at least mildly augment the effects of LV stiffening, arterial stiffening, and hypertension. These mechanisms may have meaningful effects on ventricular filling pressures with acute exercise, when there is less time for natriuretic/diuretic adjustments to help accommodate increases in stressed blood volume. This analysis does not exclude these mechanisms as contributing factors in exercise intolerance. It only suggests that these mechanisms are not primary causes of chronically elevated filling pressure. These mechanisms should still be considered in future studies of exercise intolerance, as well as in potential responses to therapeutic interventions.

Heart rate was treated as a constant in this study. Most clinical studies do not report different resting heart rates between HFpEF and non-HFpEF patients, and thus heart rate is likely not a determining factor in elevated LV EDP. However, sympathetic control of heart failure, and particularly chronotropic incompetence, may play an important role in exercise tolerance among HFpEF patients, and should be considered in future studies.

While we considered the effect of ventricular and arterial stiffening on filling pressures, arterial stiffening also increases end systolic elastance. By increasing end systolic pressure (see Fig 9A), impaired ventricular-arterial (VA) coupling may limit contractile reserve and thus limited exertional capacity [31]. Even at rest, more energy may be required to maintain normal stroke work [71]. This could contribute to cardiac ischemia, ROS, cellular stress, and fibrosis

that over time contribute to myocardial stiffening. However, these potential downstream effects of LV stiffening were not evaluated here.

This model does not attempt to represent the molecular and cellular mechanisms that govern contraction and relaxation, and thus contraction-relaxation coupling is not enforced. The activation signal is prescribed by Eq 8, and the strength of contraction is prescribed by a single parameter. This allows relaxation time and contractile strength to be altered separately so that the hemodynamic consequences each can be determined. But these mechanisms are likely not independent in reality.

While this analysis provides further support for elevated LV stiffness as the primary mechanism of HFpEF, it does not address the underlying factors that lead to elevation in LV stiffness. Understanding why LV stiffness becomes increased, and how to prevent it, may be key in preventing development of HFpEF.

Lastly, while the states of HFpEF generated here are consistent with clinical definitions of HFpEF and progress in ways consistent with clinical observations, the behavior of these HFpEF virtual patients in response to perturbations (e.g. therapeutic interventions) should be tested against clinical data. This is necessary next step before using this model to make predictions.

## Conclusions

This modeling analysis shows the important role of LV stiffness, coupled with myocardial strain as the signal for outward myocardial remodeling, in explaining the distinctly different remodeling patterns in HFpEF vs. HFrEF. It suggests that LV stiffening is both a critical contributing factor to elevation of cardiac filling pressures in HFpEF, as well as the mechanism by which outremodeling is limited and EF remains normal, since a stiffer myocardium experiences less strain and thus less outward dilatation at a given filling pressure. It supports the concept that heterogeneity within the HFpEF population is a consequence of one or more cardiac or vascular insults occurring concominant with or following myocardial stiffening. Because increased myocardial stiffness contributes to increased filling pressures and puts the patient closer to the levels needed to become symptomatic, a range of additional cardiac or vascular insults, and/or their combination, may be sufficient to push filling pressures into symtomatic levels, but not sufficient to stretch the stiff myocardium enough to trigger dilatation.

## Supporting information

**S1 Text. Full model equations.**
(DOCX)

**S2 Text. Technical Implementation.**
(DOCX)

**S3 Text. Pressure-Stress-Strain relationships in spherical thick-walled strain-stiffening vessels.**
(DOCX)

**S1 Table. Cardiac Model Parameters.**
(DOCX)

**S2 Table. Circulatory Model Parameters.**
(DOCX)

**S3 Table. Renal Model Parameters.**
(DOCX)

**S4 Table. Regulatory mechanisms model parameters.**
(DOCX)

**S5 Table. Renin Angiotensin Aldosterone System model parameters.**
(DOCX)

**S6 Table. Model Initial Conditions.**
(DOCX)

## Author Contributions

**Conceptualization:** Sanchita Basu, Jonathan R. Murrow, K. Melissa Hallow.

**Formal analysis:** Sanchita Basu, K. Melissa Hallow.

**Methodology:** Sanchita Basu, K. Melissa Hallow.

**Software:** Sanchita Basu, Hongtao Yu, K. Melissa Hallow.

**Supervision:** Jonathan R. Murrow, K. Melissa Hallow.

**Validation:** Sanchita Basu, Hongtao Yu.

**Writing – original draft:** Sanchita Basu, K. Melissa Hallow.

**Writing – review & editing:** Hongtao Yu, Jonathan R. Murrow.

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
