## [Decision Letter · Decision Letter 0]

16 Jul 2023

Dear Dr. Hallow,

Thank you very much for submitting your manuscript "Understanding heterogeneous mechanisms of heart failure with preserved ejection fraction through integrative cardiorenal mathematical modeling" for consideration at PLOS Computational Biology.

As with all papers reviewed by the journal, your manuscript was reviewed by members of the editorial board and by several independent reviewers. In light of the reviews (below this email), we would like to invite the resubmission of a significantly-revised version that takes into account the reviewers' comments.

I apologize for the slow review. The three reviews were obtained within 18 days of the paper being assigned to me.

As you can see, the reviewers had many positive comments about your work. R1 was the most favorable while R2 and R3 provided several detailed suggestions that could improve the impact of your work. I have read your paper and agree with their assessments.

Every substantiative point should be addressed in your response but I particularly encourage you to:

1) shorten and tighten the discussion.

2) provide more details on how the simulations were run (and what "simulating for 1 year" actually means)

3) review the methodology for your sensitivity analyses

We cannot make any decision about publication until we have seen the revised manuscript and your response to the reviewers' comments. Your revised manuscript is also likely to be sent to reviewers for further evaluation.

Sincerely,

Kenneth S Campbell

Guest Editor

PLOS Computational Biology

Mark Alber

Section Editor

PLOS Computational Biology

I apologize for the slow review. The three reviews were obtained within 18 days of the paper being assigned to me.

As you can see, the reviewers had many positive comments about your work. R1 was the most favorable while R2 and R3 provided several detailed suggestions that could improve the impact of your work. I have read your paper and agree with their assessments.

Every substantiative point should be addressed in your response but I particularly encourage you to:

1) shorten and tighten the discussion.

2) provide more details on how the simulations were run (and what "simulating for 1 year" actually means)

3) review the methodology for your sensitivity analyses

Reviewer's Responses to Questions

**Comments to the Authors:**

Reviewer #1: In this manuscript, the authors describe a series of simulations made in order to examine factors that produce heart failure with preserved ejection fraction (HFpEF). The authors sought to leverage this integrative model to understand the diverse manifestations of HFpEF, and in particular its relationship with the other main heart-failure modality, HFrEF. They ultimately propose a unifying hypothesis of a strain-based remodeling signal as being the key mediator of the HFpEF vs. HFrEF paradigm.

This manuscript is well-written and carefully reasoned. It offers an interesting a meaningful perspective by employing an integrative cardiovascular model in a series of strategically constructed simulations. I have only a few comments, which I believe are important, but which should be straightforward to address.

1. The mathematical definition of HFpEF did not entirely make sense. Certainly, elevated LVEDP and EF >50% are necessary and logical, but would it not also be important to manifest a drop in cardiac output? This is, after all, why heart failure matters. In Figure 5, CO is examined but perhaps should be given greater importance in the overall picture? This matter is somewhat more critical in the context of the later simulations on remodeling. In the parameter regime used to illustrate maintained HFpEF in spite of the potential for LV remodeling, does this yield a meaningful reduction of CO (such that this virtual patient would have noticeable heart failure)?

2. As a specific way of addressing the above comment, would it be possible to add CO to Figure 7, just to show that there is significant reduction in CO for both HFpEF and HFrEF outcomes?

Minor Comments:

1. I believe that in the introduction, line 14, the word ‘population’ is extraneous.

2. Under ‘Software Implementation’, I believe that the article (“a”) preceding R v4.1.1 is unnecessary.

Reviewer #2: The manuscript, PCOMPBIOL-D-23-00308, “Understanding heterogeneous mechanisms of heart failure with preserved ejection fraction through integrative cardiorenal mathematical modeling” by Basu et al., seeks to use computational modeling to determine whether one or many insults lead to heart failure with preserved ejection fraction (HFpEF). The authors find that passive stiffness is an important modulator but a ‘second hit’ is required to obtain elevated LV filling pressures (in contrast to hypertrophic patients). More than one mechanism can act as this hit.

Overall, the model provides a result that can be of substantial interest to both pre-clinical and clinical researchers working to understand HFpEF.

MAJOR COMMENTS:

1. Discussion. The discussion is overall long and difficult to follow.

1a. The discussion (and possibly the introduction) may be reduced or revised. I.e. there appear to be repetitive statements about the HFpEF phenotype. Potentially repetitive statements about the results (multiple references to what was or was not related to increased LVEDP.

1b. Could the new equations (Eq.17-21) instead be included in the supplemental materials?

1c. The most pressing comment from this reviewer is that many of the comments on the heterogeneity of HFpEF are written with a different language than the multiple components of the model- as if the model aspects and physiology were written separately. i.e. this reviewer was unable to easily identify that, because multiple interventions WITH increased LV stiffness produced increased LVEDP, a heterogeneous phenotype COULD lead to HFpEF symptoms as defined by the authors.

2. Methods- Eq.8. There are two concerns related to this formulation:

2a. This formulation is different than what is found in Bovendered (ref 47). If it is adapted in prior work from the authors, please reference it here.

2b. The formulation also does not include contraction-relaxation coupling effects. The lack of contraction-relaxation coupling may cause a substantial difference from real-physiologic changes, especially when small or moderate changes in systolic function are trialed.

3a. Methods-Eq.9. Relaxation is defined using a time, not a relaxation rate (tau) as typically defined clinically and in research literature. This difference should be noted and/or the authors should refer to this specifically as IVRT, not as relaxation.

3b. It is also concerning that the authors used a maximal tr up to 432 ms greater than 3 times the normal reported values (85-160 ms). It is unclear to this reviewer how this impacts the sensitivity analysis- could the choices of delta_tf (0.0, 0.5,1.3) impact the sensitivity analysis?

4. Discussion. A major limitation is the lack of validation to a physiologic dataset. While the authors do rightly reference prior validation of the models, comparison of these model results to clinical or pre-clinical research data (preferably more recent than the Grossman ref), would be ideal. A comment about this in the text is recommended.

5. Discussion. Another limitation is that the model does not appear to be able to remodel (i.e. Eq.2 is based on a single geometry). A comment about how remodeling (both in shape and/or mesostructured (cell/sarcomere alignment)) might impact the reported result is recommended.

MINOR COMMENTS:

6. Introduction-paragraph 3 (near refs 11-21). While the introduction includes a general summary of results of clinical trials, the authors should include a statement the HFA and ESC recommendations for HFpEF diagnosis (PMID: 31504452).

7. Results-Figure 8. The authors include increased stiffness with normal systolic function. It is likely that if collagen stiffness is increased, contractility will be reduced. Doe the authors have mathematical evidence of this? Should this be addressed in the text?

8. This reviewer would appreciate consistent or clear formatting (indent or spacing) between paragraphs.

Reviewer #3: The authors have used a computational model that couples cardiovascular and renal function to investigate various mechanisms associated with HFpEF. The following comments are submitted for the authors’ consideration.

1) There are several places in the manuscript, including the introduction and discussion, where a sentence was left incomplete. For example: “It also increases with inner chamber radius ri and decreases as wall thickness (difference between inner and outer radii, ri and ro).” and “…valvular disease predisposes one to development of HF, and different combinations of comorbidities are likely responsible for (9).”

2) The description of the parameter Δtf used in equation 9 is unclear in the text and seems to be inconsistent with how it is described in table 1. In the text, it says: “To evaluate the effect of slowed relaxation, the parameter Δtf was increased from 0 to 1.3, which increased IVRT from its baseline value of 80 ms up to 432 ms.” This makes it seem unitless and also would not increase the value of t_twitch to the range described. Table 1 says that Δtf has units of seconds and was used to increase t_twitch to +0-325% of its normal value. If this is the case, then even the percent ranges given in the table are inconsistent with the values of 80 ms up to 432 ms. This should be clarified and made consistent between the text and table.

3) In the text it states that the model was simulated for 60 days. Is this how long that simulation took to run, or does this mean you simulated 60 days of function, which would be 5.2 million heart beats. This needs to be clarified, i.e., how many heart beats were simulated and how long did it take. In addition, what resources were used to run the simulations, desktop with 8 cores, cluster with 128 cores, etc.

4) In the description of the stress driven remodeling law (model 1), why was growth assumed to be permanent and not reversible?

5) In the simulation procedure section, it says the simulation was run for 1 year. The comment here is similar to the above. Does this mean you simulated all of the heart beats in a year? Did you do any separation of time scales, which is quite common with growth laws. Also, were the rate constants tuned to experimental data to get the time course of growth over a year?

6) How were the parameter values inequations 15 and 16 selected? Also, for the normal case, it's more precise to say cf represents the combined effects of the myocytes and ecm, not that the stiffness of the myocyte and collagen are the same. They are very different.

7) For the sensitivity study, how many points were used in the parameter space? Were they evenly distributed or random?

8) I am a big fan of Sobol sensitivity indices. But one must take care in using them. In figure 3, it can be seen that there are first order sensitivity indices that are larger than the total. This is not a realistic result. If you have 3 parameters, then Stotal = S1+S12+S13+S123, i.e., it is the sum of the first and higher order indices. However, it has been shown that when doing numerical approximations of the indices, with something like a monte carlo approach, you can get artifact if you do not sample enough times. Thus, comment 7 is very important in this context. How many points did you sample in the parameter space. If you did not use enough, then your indices could be errant. The authors need to explain in detail how the study was conducted and possibly consider simulating more points in the parameter space. If simulations are too long, you could consider building a segregate model.

9) With Sobol indices, a lot of times the sampling variability in the sensitivity indices is assessed with confidence intervals. This is done by resampling the parameter space multiple times and calculating the new values for the sensitivity indices. However, in figure 3 standard deviation bars are presented. How did you assess the standard deviation? How many times were the parameters resampled?

10) Be careful using the word significant if you have not conducted a statistical analysis.

11) For the results presented in figure 8, it is unclear exactly how the different stiffness parameters were altered. Were the beta values increased for the myocytes and then the ecm? One would assume so, since the cf parameter is in the exponential and would drastically alter stiffness. Additionally, it is unclear what the parameter values are for the myocytes and ecm. They should not be the same, since these constituents behave very differently. Since the stress-stretch relations have the exact same functional form, if you did use the same values for both, then perturbing each one would lead to the same outcome and not distinguish the influence of each correctly. This should be stated clearly.

12) No limitations are given in the manuscript. These should be noted in a subsection in the discussion.

**Have the authors made all data and (if applicable) computational code underlying the findings in their manuscript fully available?**

Reviewer #1: Yes

Reviewer #2: Yes

Reviewer #3: **No: **The manuscript and supplement provide extensive details about the equations that were employed, but the code itself is not available.

PLOS authors have the option to publish the peer review history of their article (what does this mean?). If published, this will include your full peer review and any attached files.

Reviewer #1: No

Reviewer #2: No

Reviewer #3: No

Figure Files:

Data Requirements:

Please note that, as a condition of publication, PLOS' data policy requires that you make available all data used to draw the conclusions outlined in your manuscript. Data must be deposited in an appropriate repository, included within the body of the manuscript, or uploaded as supporting information. This includes all numerical values that were used to generate graphs, histograms etc.. For an example in PLOS Biology see here: http://www.plosbiology.org/article/info:doi%2F10.1371%2Fjournal.pbio.1001908#s5.
---

## [Decision Letter · Decision Letter 1]

13 Oct 2023

Dear Dr. Hallow,

We are pleased to inform you that your manuscript 'Understanding heterogeneous mechanisms of heart failure with preserved ejection fraction through cardiorenal mathematical modeling' has been provisionally accepted for publication in PLOS Computational Biology.

Best regards,

Kenneth S Campbell

Guest Editor

PLOS Computational Biology

Mark Alber

Section Editor

PLOS Computational Biology

Reviewer's Responses to Questions

**Comments to the Authors:**

Reviewer #1: The authors have effectively addressed my concerns and I find the paper substantially improved.

Reviewer #2: The revised manuscript PCOMPBIOL-D-23-00308R, “Understanding heterogeneous mechanisms of heart failure with preserved ejection fraction through cardiorenal mathematical modeling” by Basu et al. uses computational modeling to better understand HFpEF.

The computational model includes adaptable components including cardiac function and structure, cardiovascular flow, and renal homeostasis mechanisms. The authors provide evidence that increased myocardial stiffness with at least one additional stress can lead to HFpEF like conditions, including increased stiffness.

Overall, the revision addresses this reviewer’s major concerns.

Reviewer #3: The authors have done well to address my comments, particularly those related to the Sobol Sensitivity Analysis. I appreciate the pivot to using the approach by Azzini.

**Have the authors made all data and (if applicable) computational code underlying the findings in their manuscript fully available?**

Reviewer #1: Yes

Reviewer #2: Yes

Reviewer #3: Yes

PLOS authors have the option to publish the peer review history of their article (what does this mean?). If published, this will include your full peer review and any attached files.

Reviewer #1: No

Reviewer #2: No

Reviewer #3: No

---

## [Editor Report · Acceptance letter]

1 Nov 2023

PCOMPBIOL-D-23-00308R1 

Understanding heterogeneous mechanisms of heart failure with preserved ejection fraction through cardiorenal mathematical modeling

Dear Dr Hallow,

I am pleased to inform you that your manuscript has been formally accepted for publication in PLOS Computational Biology. Your manuscript is now with our production department and you will be notified of the publication date in due course.

With kind regards,

Zsofia Freund
